# Threshold Definition for Monitoring Gapa Landslide under Large Variations in Reservoir Level Using GNSS

Shuangshuang Wu [1,2], Xinli Hu [1,*], Wenbo Zheng [3], Matteo Berti [2], Zhitian Qiao [2] and Wei Shen [2]

1   Faculty of Engineering, China University of Geosciences, Wuhan 430074, China; wshuang@cug.edu.cn
2   Department of Biological, Geological and Environmental Sciences, University of Bologna,
    40126 Bologna, Italy; matteo.berti@unibo.it (M.B.); zhitian.qiao2@unibo.it (Z.Q.); wei.shen2@unibo.it (W.S.)
3   School of Engineering, University of Northern British Columbia, Prince George, BC V2N 4Z9, Canada;
    Wenbo.Zheng@unbc.ca
*   Correspondence: huxinli@cug.edu.cn; Tel.: +86-13907152610

**Abstract:** The triggering threshold is one of the most important parameters for landslide early warning systems (EWSs) at the slope scale. In the present work, a velocity threshold is recommended for an early warning system of the Gapa landslide in Southwest China, which was reactivated by the impoundment of a large reservoir behind Jinping's first dam. Based on GNSS monitoring data over the last five years, the velocity threshold is defined by a novel method, which is implemented by the forward and reverse double moving average of time series. As the landslide deformation is strongly related to the fluctuations in reservoir water levels, a crucial water level is also defined to reduce false warnings from the velocity threshold alone. In recognition of the importance of geological evolution, the evolution process of the Gapa landslide from topping to sliding is described in this study to help to understand its behavior and predict its potential trends. Moreover, based on the improved Saito's three-stage deformation model, the warning level is set as "attention level", because the current deformation stage of the landslide is considered to be between the initial and constant stages. At present, the early warning system mainly consists of six surface displacement monitoring sites and one water level observation site. If the daily recorded velocity in each monitoring site exceeds 4 mm/d and, meanwhile, the water level is below 1820 m above sea level (asl), a warning of likely landslide deformation accelerations will be released by relevant monitoring sites. The thresholds are always discretely exceeded on about 3% of annual monitoring days, and they are most frequently exceeded in June (especially in mid-June). The thresholds provide an efficient and effective way for judging accelerations of this landslide and are verified by the current application. The work presented provides critical insights into the development of early warning systems for reservoir-induced large-scale landslides.

**Keywords:** threshold; landslide; early warning system; velocity; water level; GNSS

## 1. Introduction

Landslides pose great threats to life and property. Many large-scale landslides have now been reactivated by the impoundment of large reservoirs behind high dams in southwestern China. The heights of these reservoir impoundments reach hundreds of meters. For example, the impoundment of Jinping's first reservoir is 230 m in height, and the fluctuating water level is 267% greater than that of the well-known Three Gorges Reservoir Area (TGRA) [1]. The framework of landslide monitoring has been established to provide a scientific basis for geohazard forecasting and warning in the TGRA [2]. However, many high dams with large reservoirs are now being constructed to the west of the TGRA, where the in-situ monitoring technologies and early warning strategies have not been applied and developed immediately. Consequently, large-scale landslides can be reactivated during reservoir construction and subsequent operation and may evolve into destructive failures [3]. Thus, designing and implementing monitoring and early warning systems (EWSs)

are urgent for the safety of dam operations as well as that of residents and property on both river banks.

EWSs can be used for disaster mitigation based on landslide monitoring and prediction. Researchers have been encouraged to establish EWSs for various landslides around the world [4], because they are cost-effective and efficient. Concerning the mechanisms of landslides, scenario analysis, the evolution process, and long-term in situ monitoring laid an important foundation for the further development of EWSs. Researchers design and implement effective EWSs at regional and slope scales, respectively. They are established based on geological knowledge, field investigation, monitoring of geohazards with threats, and the choice of monitoring parameters [4–7]. For instance, an advancing regional warning model for rainfall-induced landslides, based on rainfall thresholds, has been proposed in a landslide-prone area in the Campania region, Italy [8]. An EWS has also been designed for the La Saxe rockslide with predefined displacement and/or velocity thresholds to offer the near real-time of failure [9].

There have been substantial advances in EWSs thanks to the development of modern monitoring technology. For instance, a global navigation satellite system (GNSS) has been installed on a deep-seated landslide in Slovenia to establish an EWS which has the advantages of low cost, open-source processing software, and automatic data collection over the Internet. The displacement data are correlated with rainfall data to reveal how different parts of the landslide react to precipitation and further develop the EWS [10]. Other commonly employed monitoring instruments include the ground-based interferometric synthetic aperture radar (GBInSAR), ShapeAccelArray (SAA), unmanned aerial vehicle (UAV), crack extensometers, and piezometers [11–13].

The importance of threshold definition in EWSs for landslides has been widely recognized [14]. Rainfall is the main trigger of landslides, and for this reason rainfall thresholds are the most well-known threshold definition methods in landslide and debris flows [15–17]. In contrast, water level thresholds are also used for reservoir-induced landslides, because these landslides' behaviors are strongly related to reservoir impoundment and fluctuations [18–20]. In this case, the two main landslide types are seepage-induced and buoyancy-induced, which are related to landslide geometry and materials [2]. Of course both rainfall and water levels influence the infiltration and pore water pressure within slopes, thus affecting the stability of landslides. Therefore, more parameters (e.g., soil moisture and underground water table) that directly affect the landslide body and sliding surface are needed to predict failure or establish EWSs [21–25]. In addition to various water-related thresholds, another effective threshold for EWSs at the slope scale is kinematics parameters, which can objectively demonstrate the behaviors of geohazards. Basically, measured displacement and its derivatives are most widely mined for threshold definition with the aid of useful on-site monitoring data [26–30].

A successful EWS at the slope scale encompasses proper early warning indicators, primarily using monitored displacement and its velocity and, sometimes, environmental quantities, such as critical water level and rainfall threshold [19,31,32]. The EWS model based on the three-stage creep theory of rock and soil materials is widely used in landslides. The model was first proposed by Saito in the 1960s and further developed with the popularity of the concepts of inverse velocity and displacement increment [26,33–35]. It was successfully applied on landslide forecasting with the improved accuracy of displacement increment and velocity. Systems can release alerts after observations reach default thresholds in order to save lives and properties and prevent environmental damage [36–38], but the prerequisite is that the landslide must have finished the constant deformation stage and entered the acceleration stage in Saito's model, because the velocity in the constant deformation phase needs to be predetermined for the establishment of an EWS.

On the other hand, the cumulative displacement of landslides under periodic reservoir water fluctuations always shows a constant period and some periodic (yearly or biennially) abrupt accelerations [39]. This cyclic displacement feature increases the difficulty of judging the landslide evolution stage. With this background, this study was carried out in a



riverbank landslide located in the Yalong River, one of the tributaries of the upper Yangtze River, China. The Gapa landslide, with a volume of 13 million cubic meters, was chosen as a typical case by its type, scale, and large deformation. The warning level of the EWS is determined by five-year continuous monitoring with the aid of an improved Saito's model. The displacement data are thoroughly analyzed using the running average velocity to define a threshold. Both the velocity threshold and water level threshold are recommended for the Gapa landslide's EWS.

## 2. General Setting

### 2.1. Reservoir and Landslide Features in the Study Area

The western part of the upper Yangtze River is a landslide-prone area, especially the reservoir area. Many landslides were triggered due to large periodic water level changes of reservoirs in this area, seriously affecting the safety of bank residents and hydropower facilities. Seventy-seven landslides have been found in nine reservoirs in southwestern China along the Yalong River, Jinsha River, Dadu River, and Min River (Figure 1). These landslides developed behind hydropower dams, and 97% of these previously stable landslides have been triggered or reactivated by these reservoir impoundments. Thirty-six landslides are situated on the banks of the Yalong River. Among the 77 landslides, 42 landslides have volumes of more than 10 Mm$^3$. Southwestern China is also a bedrock landslide-prone area, due to its geomorphological and lithological features. Twenty-four landslides are formed by topping or bending, and 34 landslides are Triassic, in terms of the stratigraphic age of the sliding surface. These sliding-prone lithologies of the Triassic strata are sandstone, metamorphic sandstone interspersed with shale, slate, and coal layers.

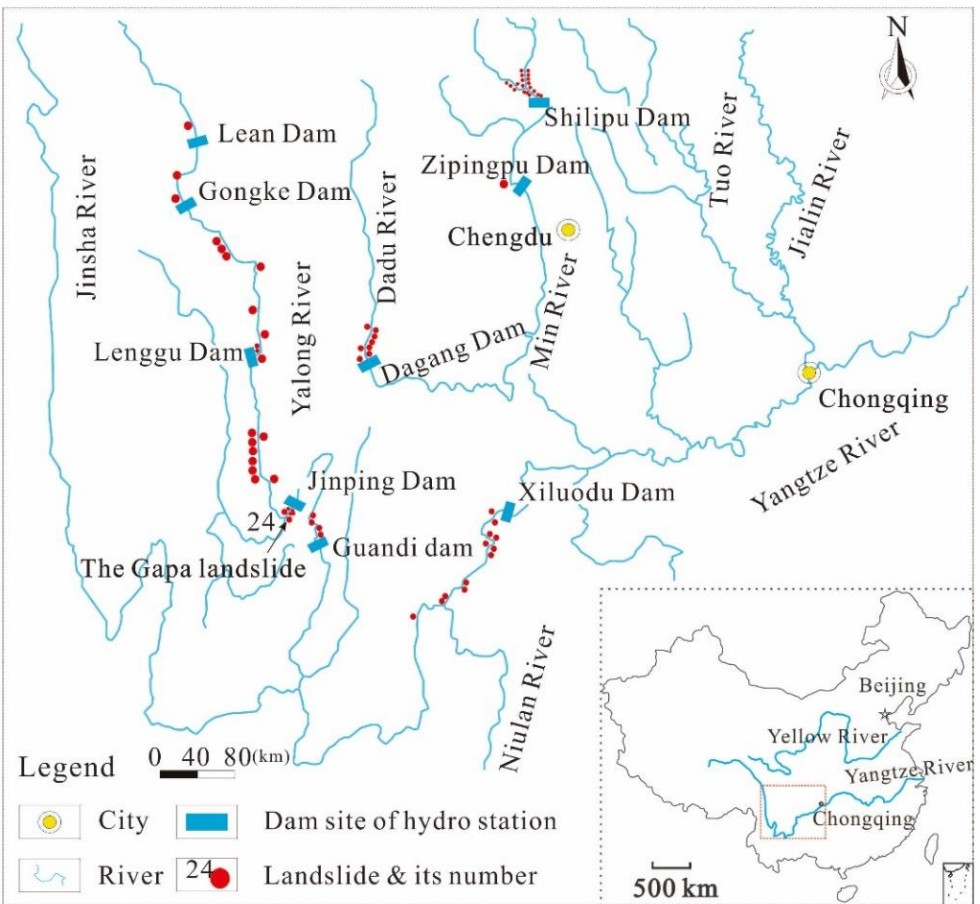

**Figure 1.** Riverbank landslide distribution in reservoir areas, Southwest China.

### 2.2. The Brief of the Gapa Landslide

The Gapa landslide is representative of the area, with almost all the main landslide features described above. The Gapa landslide is denoted as landslide no. 24 in Figure 1, behind the first Jinping hydropower dam.

Figure 2 presents the topographic map and typical cross-section of the Gapa landslide. The landslide is approximately 980 m long and, on average, 360 m wide from the front view, with a height of 470 m from the highest detected head scarp to its projected toe. The depth of the sliding surface is estimated to be 60 m. Accordingly, the estimated landslide volume is 13 Mm$^3$. The landslide consists of soil and blocky crushed rocks, mostly silty slate, mudstone, and conglomerate. It was formed by gravity in a pattern of slope bedding failures, and the bedding zone still exists today in bedrock at a certain depth below the sliding surface.

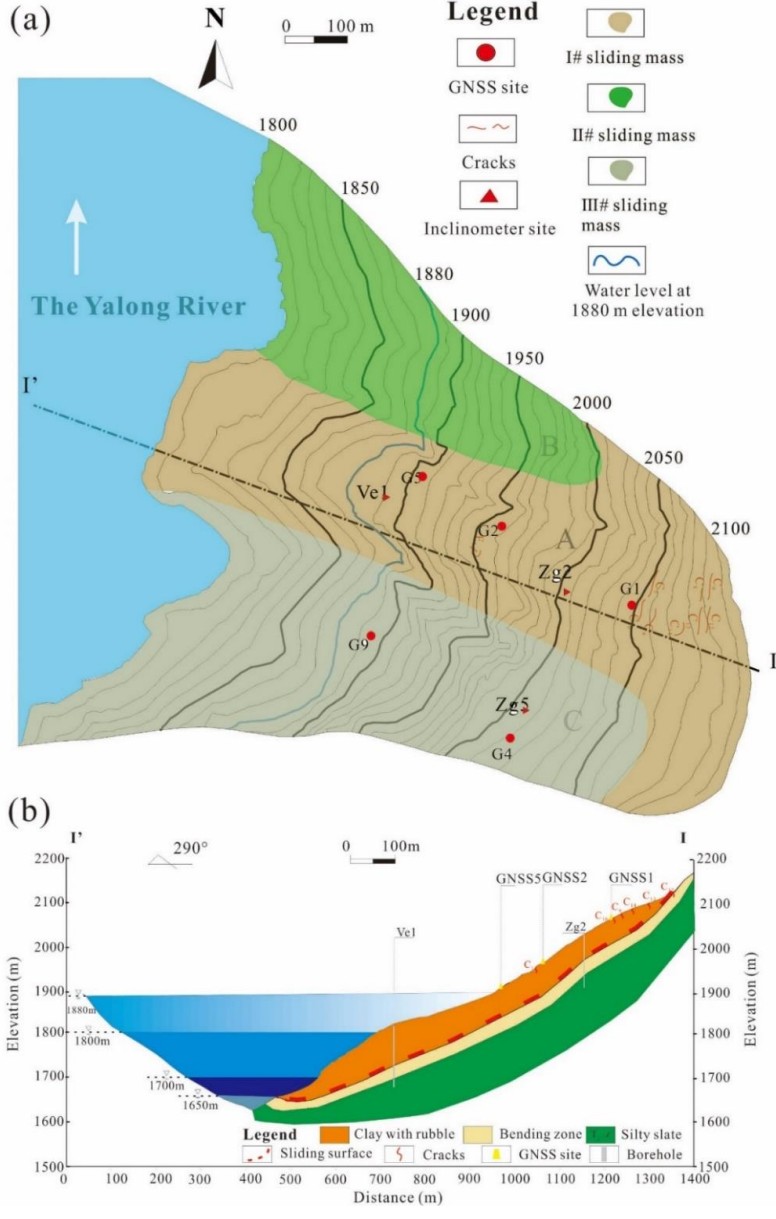

**Figure 2.** Engineering geological plan (**a**) and typical cross-section (**b**) of the Gapa landslide. (**a**) The landslide surface below 1800 m asl has been permanently submerged by the reservoir lake; (**b**) the reservoir impounded to 1880 m asl in three stages (1650–1700 m asl in November 2012, 1700–1800 m asl in July 2013, and 1800–1880 m asl in September 2014).

The first Jinping reservoir initially impounded in November 2012. The original lake level was 1650–1655 m asl, and it rapidly rose to 1700 m asl and remained at this lake level for a few months. Then, the reservoir was filled to 1800 m asl in August 2013. The reservoir impoundment in this area greatly affected the hydrogeological condition of these riverside slopes and landslides. After that the first impundment, the lake level cyclically fluctuated between 1800-m and 1840-m elevations in 2014 and between 1800-m and 1880-m elevations in 2015. The reservoir submerged half of the landslide at the lake level of 1880 m asl, as per Figure 2. These fluactuations caused continuous infiltration and discharge of reservoir water inside the slope. Local collapses occurred in the front, and cracks continuously developed and enlarged in the middle and upper parts of the landslide in 2014 and 2015. The reservoir operation recognized the need to increase our understanding of the evolutionary mechanisms and movement triggers of the Gapa landslide, as well as to implement a monitoring and early warning system for hazard control.

## 3. Evolution Process and Future Trend

### 3.1. Slope Behavior before the Impoundment

#### 3.1.1. Historical Failure

The materials at the Gapa landslide's sliding surface were collected to determine the formation age. The result of soil thermoluminescence dating indicates that the landslide was formed between 14,600 and 23,500 years ago, which is during the late Pleistocene and early Holocene. Below the sliding surface, the bedrock can be divided into three sublayers, based on the bending degree, namely, strong bending, weak bending, and normal zones (Figure 3). The different bending degrees were illustrated by drill cores, which are neatly arranged with different fracture dips and block thicknesses, providing the bending evidence.

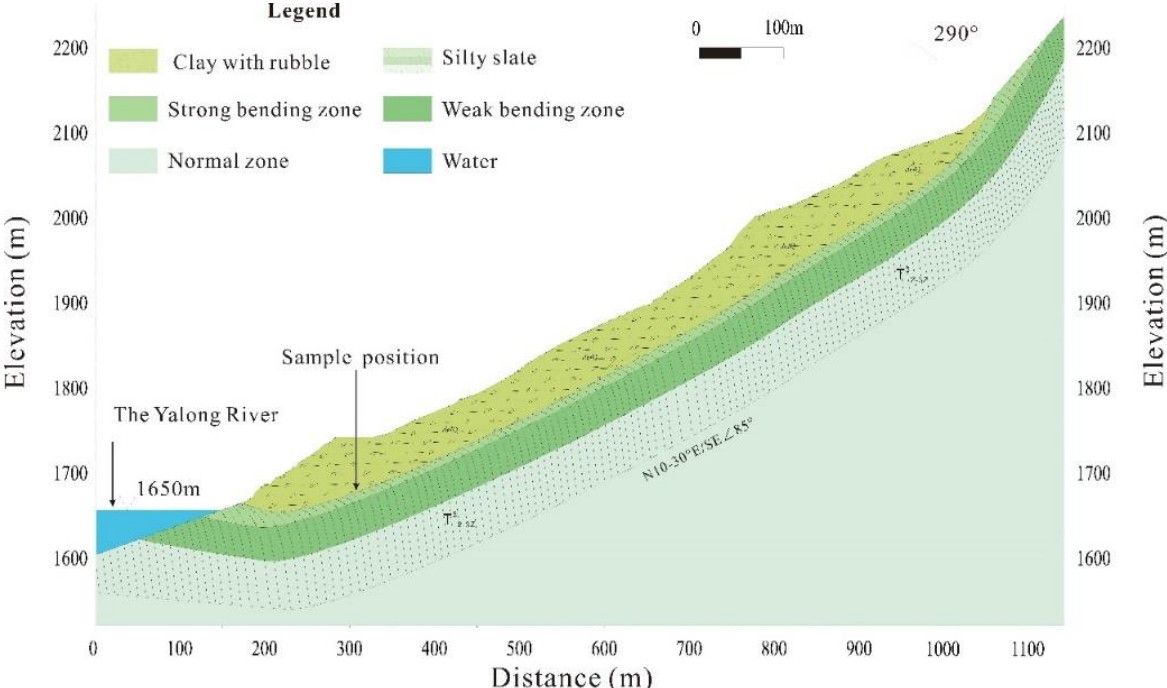

**Figure 3.** Cross-section of the Gapa landslide formed by bending pattern; the slope geometry refers to Zone A of the landslide before the reservoir impoundment.

#### 3.1.2. Slow Movement before Impoundment

After the topping failure, it is hard to confirm the movement or evolution of the landslide without records, but satellite-based images and investigation provide insights

into the landslide deformation in the 21st century. Figure 4 shows the images of the landslide in 2005 and 2011 from Google Earth. The difference in geomorphology between 2005 and 2011 is not significant. Before the reservoir impoundment (in November 2012), GPS stations were installed to monitor the deformation between 2007 and 2012. The GP2 station is near the location of the GNSS2 shown in Figure 5a; both are situated in the middle of the main cross-section, as shown in Figure 2. Figure 5b presents the cumulative displacement, with an average velocity of 14 mm/year during 2007–2012. According to the landslide velocity classification proposed by Cruden and Varnes (1996) [40], the Gapa landslide velocity before impoundment was below 15 mm/year and is considered to be very low. Without considering unknown earthquake occurrences or other influencing factors in history, it can be deduced that the Gapa landslide was formatted by rock slope topping and then deformed very slowly in the 21st century until the reservoir impoundment.

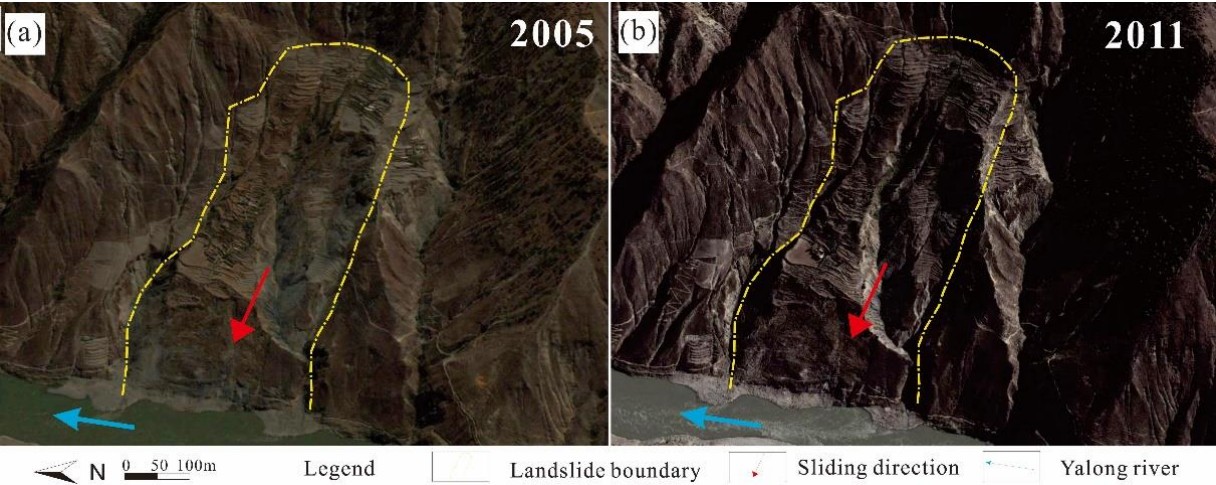

**Figure 4.** Satellite images of the Gapa landslide before reservoir impoundment (**a**) in 2005 and (**b**) in 2011 (Source: Google Earth).

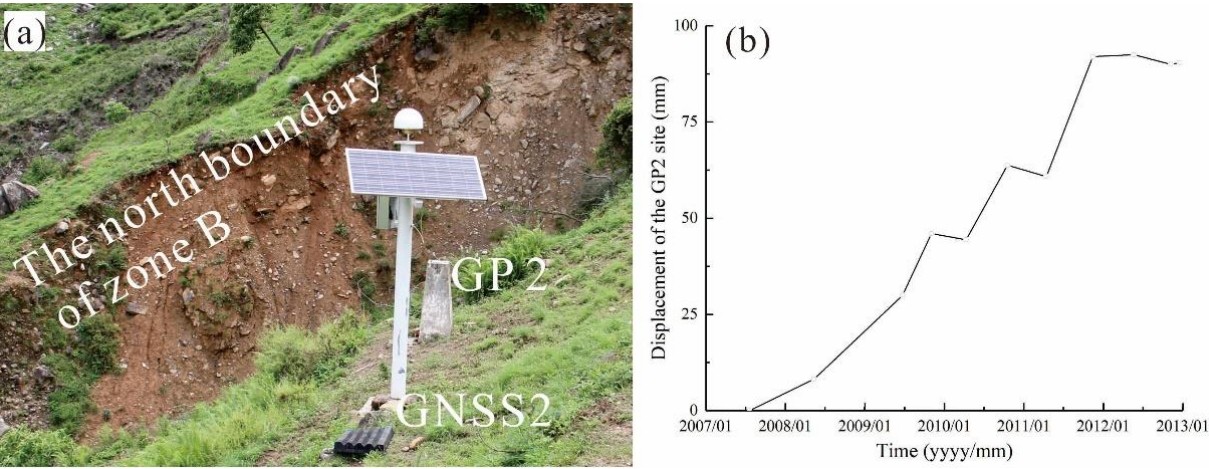

**Figure 5.** (**a**) GNSS2 unit and GP2 unit and their relative positions; (**b**) the cumulative displacement curve of GP2 between 2007 and 2012.

### 3.2. Recent Deformation Subjected to Impoundment

After the reservoir impoundment was completed, the Gapa landslide was expected to reactivate, due to the influence of reservoir water. Surface cracks developed on the ground highlighting the boundary of the landslide. Collapses occurred nearly above the highest

water level. Consequently, field investigations were carried out to assess the landslide activity, and monitoring instruments were installed partially in 2016 and 2018 to set up a deformation monitoring system. The layout of the landslide monitoring is also shown in Figure 6. The monitoring plan includes three subsurface monitoring sites, six surface monitoring points, and an unmanned aerial vehicle (UAV).

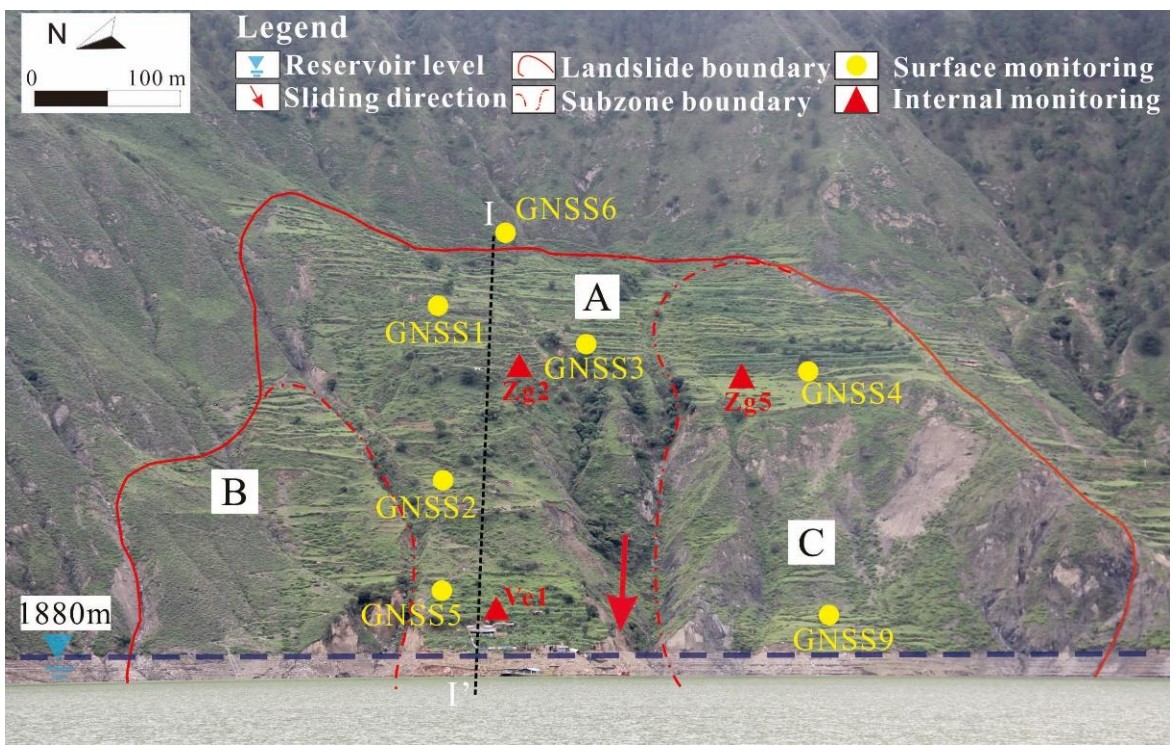

**Figure 6.** Overview and the monitoring layout of the current Gapa landslide.

The subsurface displacement was monthly monitored by inclinometers. Because the large deformation is beyond the allowable value of the instrument, none of the inclinometers can constantly access data, and they have been disused. However, the depth of the sliding surface was revealed by the available inclinometer measurements.

The crack distribution, using photos from the UAV with a camera, is illustrated in Figure 7. From the front view, the main boundary and zoning are highlighted with the red line, with a length of nearly 1 km and an average width of 2 m. In addition, two parallel tension cracks are on the upper part of the landslide at 2100 m and 2060 m asl, running through Zone B to Zones A and C. A transverse crack extends from Zone B to Zone A at 1900 m asl which does not pass to Zone C, due to the right-side gully and cliffs. Meanwhile, the UAV image shows the detailed cracks, which are rendered by solid yellow lines (Figure 7). In addition to those presented in the image, more than 30 visible cracks in total were found during the survey in 2018 with a certain distribution law. Most of the cracks continue to develop according to the later investigations in 2019 and 2020.

According to Kilburn and Petley (2003) and Xu et al. (2008), cracks on the ground surface are likely to gradually form a complete crack system with the increase of landslide deformation in space [41]. Unless the influences of external factors diminish, the cracking can grow at an accelerating rate until they coalesce into a major plane of failure; additionally, the cracking is enhanced by circulating water for deep-seated movements [42].

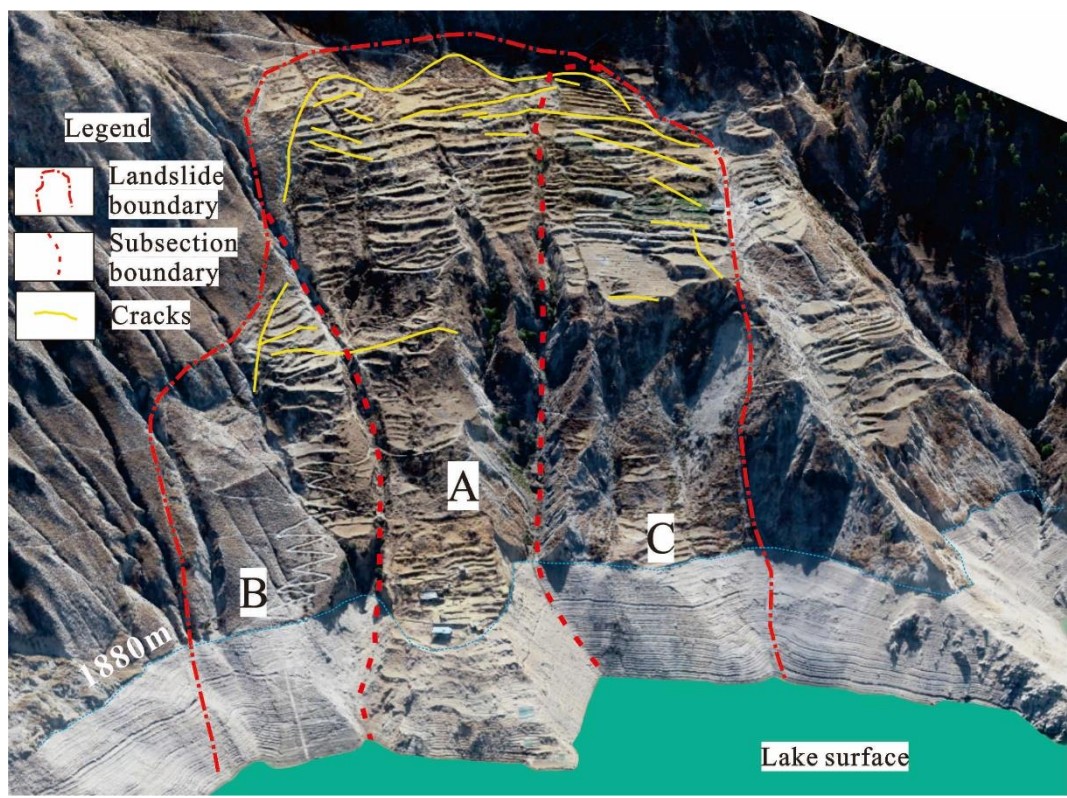

**Figure 7.** Main cracks of the overall Gapa landslide; this UAV image was captured in March 2019.

The surface displacements are daily recorded and obtained by a global navigation satellite system (GNSS). The GNSS processing is performed automatically in a server. Each unit sends raw GNSS data to the base station via a 3G signal. The base station is 11 km away from the landslide site, located in the dam safety center. GNSS data were processed in baseline mode by a batch least squares adjustment approach [10]. The observation was weighted with the inverse of the sine of the satellite elevation; its rate is 30 s, and the interval of the estimated coordinates is 24 h. However, the estimated daily displacement time series are formatted by the Yalong River Hydropower Development Company, which means that the company has direct access to the data, instead of every end-user.

The GNSS data between January 2016 and September 2020 were provided by the company for this study (over five years). The monitored main displacement is horizontal along the east-west, out of more than 60% of the combined displacement [43]. To check their validity, we reset one reference point on a supposedly stable terrain outside the landslide boundary (GNSS6, hereafter G6). The displacement data from the G6 site span from −8.5959 to 8.3742 mm between 2018 and 2020, converging on ±1 cm.

The surface horizontal displacement monitoring results are presented in Figure 8. During the first fluctuation cycle of the reservoir lake, the landslide deformation was unexpected and, therefore, was not monitored. The fluctuation cycle here includes different periods of the reservoir drawdown (Jan. to Jun.), filling (Jul. and Aug.), and maintaining at the highest level (Sep. to Dec.) in each year. From the second to the sixth fluctuation cycle, the movement of the landslide was daily recorded. The cumulative displacement increased during by the reservoir water fluctuations between 2016 and 2020. Some accelerated deformations are highlighted on the displacement curves during each fluctuation except the sixth fluctuation, whose lowest water level (1812 m elevation asl) is more than 10 m higher than usual, due to excessive rainfall. The acceleration always happened during lower water levels. Since the phase under a lower water level is short, the acceleration disappears after the water level increases.

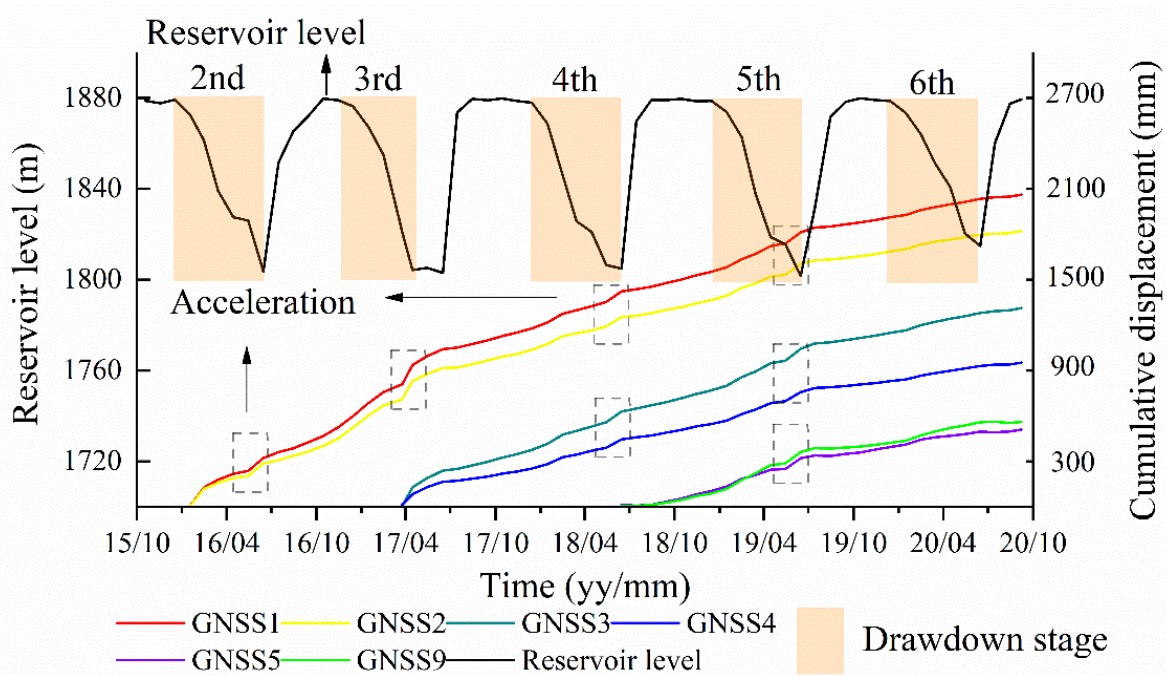

**Figure 8.** The surface displacement of GNSS units and water level from the 2nd to 6th fluctuation of the reservoir. Obvious accelerations indicated by the dotted box occur when the reservoir water drops to the lowest level.

Wu et al. (2021, 2022) and Hu et al. (2021) provided additional details about the impacts of reservoir water and its fluctuations on yearly periodic accelerations of the Gapa landslide [43–45]. The accelerations during the reservoir drawdown are mainly induced by: (1) the release of the buttressing effect of the reservoir lake; and (2) the outward seepage force inside the landslide before the dissipation of pore water pressure. Moreover, reservoir water dropping to the lowest level occurred within the local rainy season, which may also enlarge the acceleration behaviors [45]. More detailed displacement features are illustrated in Section 4.

### 3.3. Projected Displacement Trend

To investigate the evolution of the Gapa landslide, the displacement vs. time curve is plotted in order to explain the observed evolution stages (Figure 9). Five stages, including two possible displacement trends, are shown in the landslide lifespan curve: (1) topping failure (AB), which occurred during the late Pleistocene and early Holocene; (2) very slow motion (BC) until the reservoir impoundment in 2012; (3) new deformation (CD) caused by reservoir impoundment and fluctuations between 2013 to 2020. Notably, the interannual deformation rates in stages 2 and 3 have been compared by the typical monitoring sites GP2/GNSS2 (Figure 9). The velocities during stage 3 are dozens of times higher than during stage 2, indicating the reactivation of the Gapa landslide, but the velocity yearly or biennially decreases, instead of increasing, during the second and sixth fluctuations, which seems to be consistent with the features of the initial deformation phase based on the three-stage creep model (Satio 1969 [33]). In other words, if taking point C as the starting point of the old Gapa landslide reactivation, the displacement behavior of the reactivated Gapa landslide is unlikely to be the same as the Qianjiangping landslide or the Vajont landslide, which immediately collapsed after impoundment or the first reservoir water drawdown [46,47]. According to the Saito's model, the landslide has exhibited the initial stage. Based on this conceptual framework, predictions can be made about landslide deformation under the next long-term periodic fluctuations.

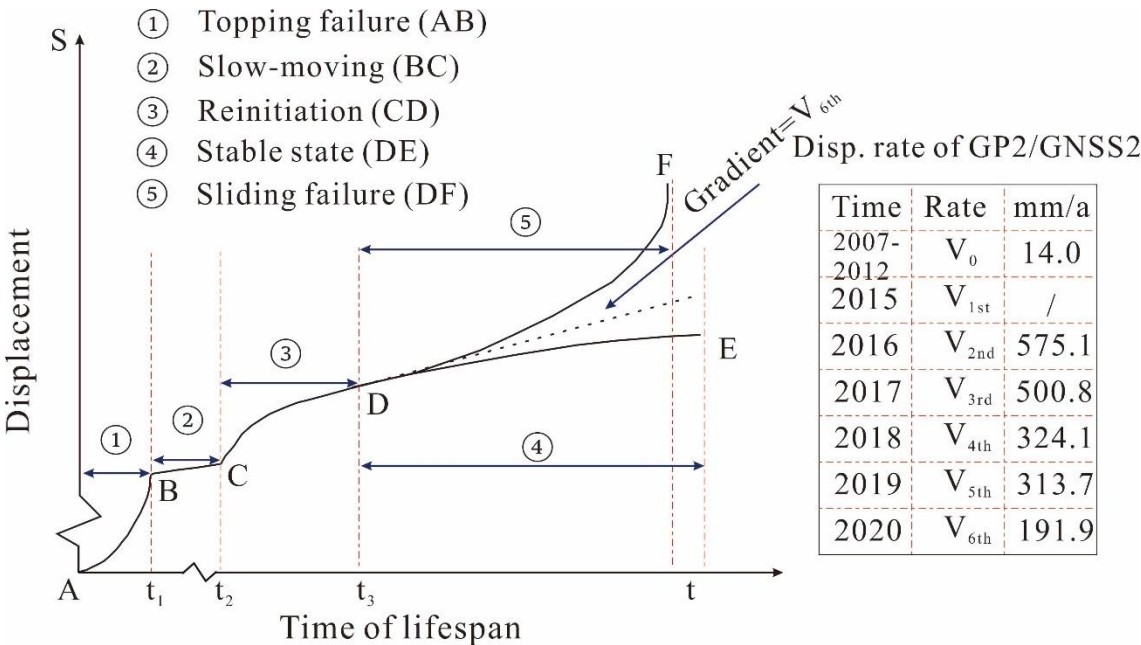

**Figure 9.** Displacement vs. lifespan curve of the Gapa landslide. In this figure, $t_3$ is the present time; therefore, stages AB, BC, and CD are supported by data, while stages DE or DF are inferred. The table on the right summarizes the yearly velocity of the GP2/GNSS2 unit between 2007 and 2020, indicating movement spans from constant rate, acceleration, and deceleration and corresponding to some period in stages BC and CD.

Thus, in addition to these three known stages, two likely stages—(4) stable state (DE) and (5) sliding failure (DF)—are suggested. Stage 4 means that the landslide can gradually adapt to the reservoir water influences after experiencing perennial deformations and finally recovers to the slow-moving state or reaches the stable state. Conversely, the reservoir water may lead to cumulative strain effects, reducing the mechanical strength within the landslide body. In this case, the Gapa landslide could experience a constant deformation stage and then acceleration deformation, as stage 5 illustrates.

At present, considering that the monitoring period is relatively short and thus has limited representativeness, the behavior of the landslide could change substantially in the following years. However, other factors, such as unexpected heavy rain, could jointly affect the landslide in the future. Based upon the above uncertainties, an EWS with a degree of security is strongly recommended, as well as effective thresholds.

## 4. Threshold Definition

### 4.1. Warning Model and Warning Level Determination

The modified creep model described above was used to develop the Gapa landslide early warning systems (EWSs) [36,37,48]. As shown in Figure 10, deformation before failure from the initial to acceleration stages can properly correspond to a four-class warning in the order of "attention level", "caution level", "vigilance level", and "alarm level". This EWS model can be established using in situ displacement monitoring. The data of displacement-time series can be presented in this way to judge the deformation stage of the monitored object and help to set the warning level, but the starting point of the monitoring time is often not at the onset of deformation [7]. Thus, this model cannot be realized without an understanding of the deformation behaviors and history, as well as of the evolution process, of the landslide unless the landslide has definitively entered the accelerated deformation stage.

Based on existing literature, the movement of the Gapa landslide is based on the motion of the deep-seated sliding surface [43]. Therefore, the EWS for the Gapa landslide under the periodic reservoir level fluctuations has now been preliminarily developed based

on: (1) the knowledge of the landslide evolution process from topping to sliding; and (2) daily surface displacement data using GNSS monitoring over the last five years, as the motion at depths is close to that of the ground.

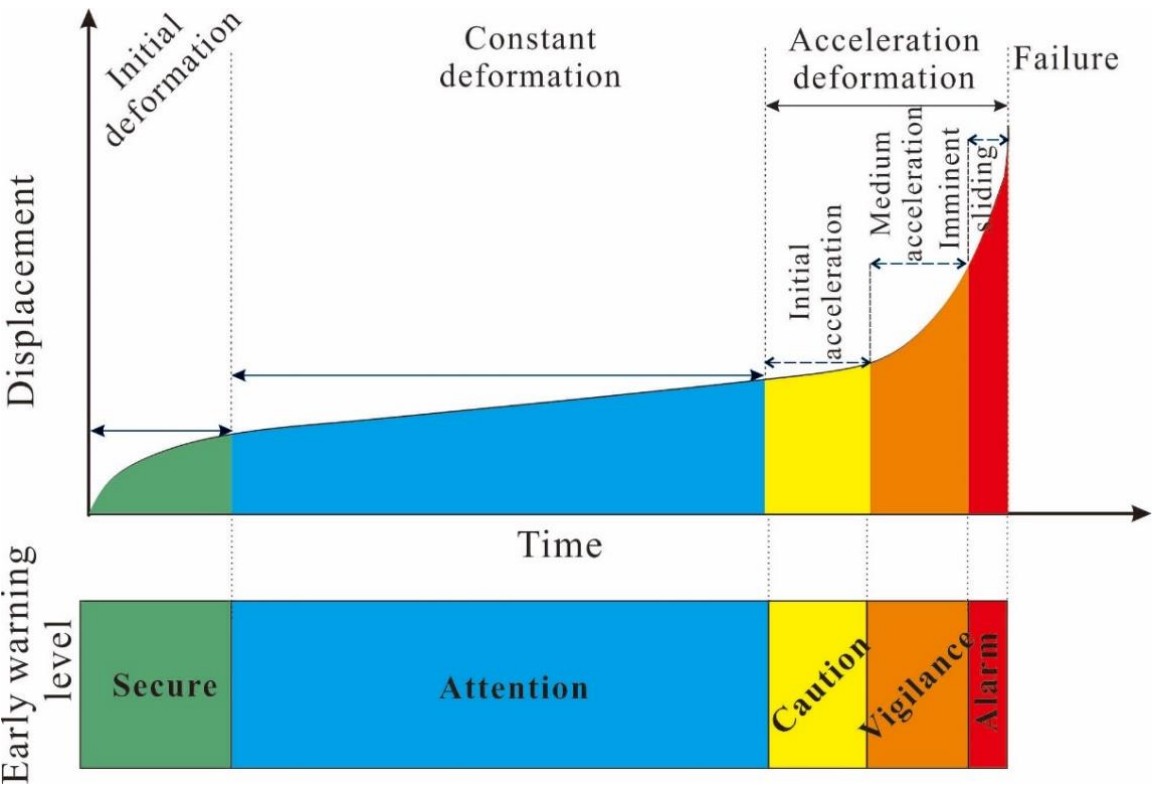

**Figure 10.** The warning model with four warning levels using the improved Saito's model (modified from [36]).

As stated above, the yearly rate of the landslide during the second and sixth fluctuations decreased by 40% (Figure 9), indicating a trend that the landslide movement now is experiencing the initial deformation stage and may enter the consistent deformation stage. However, it is hard to determine the ranges of deformation rates during the next couple of fluctuations or distinct futures, as this type of movement exhibited rapid acceleration during every reservoir drawdown period (Figure 8). Meanwhile, according to the literature, some reservoir landslides in Southwest China with a longer monitoring period have the adaptability or resistance to undergo the same scenario after they have deformed under the same and periodic fluctuations [49–51]. It is more likely that the deformation rate during the next couple of fluctuations will not exceed the velocity that occurred during the sixth fluctuation under the same single scenario, despite the uncertainties mentioned above. Therefore, it can be concluded that the level of warning is now properly releasing the attention level in terms of safety, as illustrated in Figure 10. The objectives of monitoring at the attention level are to determine whether acceleration is occurring or not, whether the acceleration phase is continuous, and whether it leads to tertiary deformation. Thresholds should, therefore, be defined at this level to issue warnings that acceleration has occurred or is likely to continue.

### 4.2. Velocity Threshold Based on Moving Average

The threshold must be conservative and minimize unnecessary warnings [52]. For this purpose, the displacement data are further analyzed to define the optimal threshold levels. Note that in this EWS model, the velocity, instead of the accumulative displacement, is applied to design the threshold, because the accumulative displacement can be easily

influenced by the landslide scale and failure mode. Velocity values can be computed by original records of displacement according to the following equation:

$$v_j = \frac{S_j - S_{j-1}}{t_j - t_{j-1}} \tag{1}$$

where $v$ is the displacement rate recorded between $j$ and $j - 1$ readings, $S$ is the displacement recorded at reading $j$, and $t_j$ is the date, expressed as a number corresponding to reading $j$. Note that the absence of daily displacement data is supplemented by linear interpolation.

The velocity vs. time series is illustrated in Figure 11, taking GNSS2 (G2) as an example. As gray scatters are shown in the figure, two features can be found: (1) the velocity points are chaotic, which hides the order or rule of the changes in velocity and time; and (2) many velocity points are at noise level and some below the zero line. The second feature is frequently observed in field monitoring data but without real meaning. Obviously, no thresholds can be directly picked by the gray scatters, but the two defects can be overcome by introducing the moving average (MA) method to deal with the original data, which is a useful method and commonly applied to data smoothing and decomposing in landslide fields [11,45,53,54]. The moving average displacement velocity can be calculated by the original time series and can be expressed as:

$$v'_t = \frac{v_t + v_{t-1} + \ldots + v_{(t-n+1)}}{n}, (t \geq n) \tag{2}$$

$$v''_t = \frac{v'_t + v'_{t-1} + \ldots + v'_{(t-n+1)}}{n}, (t \geq 2n - 1) \tag{3}$$

where $v_t$ is the recorded data of the original time series at time $t$, $v'_t$ is the value of the time series at time $t$ by first moving average, $v''_t$ is the value of the time series at time $t$ by second moving average, and $n$ is the moving interval for generating each data point, also known as moving window. Note that the moving window is the most important parameter in determining the MA trend, instead of the setting of the onset or the end of the time series. For example, $n$ can be set between days and years to represent the short- and long-term trends of the velocity.

The seven-day running average is used once (Equation (2)) and twice (Equation (3)) to plot G2's velocity over short-term durations. Shown as black and orange scatters, most of the scatters distribute on the positive zone above the $-1.3$ and $-0.2$ mm/d delimited lines, respectively. Meanwhile, the change trends in velocity are obvious, with some accelerations and decelerations, which is beneficial for finding the most common rates and maximum rates.

Based on MA, short- and long-term G2 velocity trends are summarized in time series, as shown in Figure 12. The running values are chosen as 7, 30, 120, and 360 days to represent weekly, monthly, seasonally, and yearly velocity trends of G2, respectively. As the running value increases, the velocity curve smooths and the maximum rate decreases. Meanwhile, the maximum rates both in the short and long terms reduce as the fluctuation time increases over the monitoring period. For instance, the weekly maximum rate in 2016 and 2017 is 8.2 mm/d, then declines to 3.5 mm/d in 2018 and 2019, and further decays to 2 mm/d in 2020 (gray line). This finding can be drawn from the monthly maximum rate as well (blue line). From 2016 to 2020, while the number of fluctuations in the reservoir increases, the induced landslide deformation rate gradually decreases. The MA results are consistent with the previous observation (Figures 8 and 9), which is important for determining the threshold.

Meanwhile, the accessed data should be processed in a timely manner in an EWS, notably the weight of new data. An EWS cannot rely on the long-term MA, otherwise the time effectiveness of the EWS will be missing. Thus, the weekly running, which presents the short-term velocity trend, seems to be more suitable for finding a threshold than others.

Meanwhile, in order to make the seven-day running velocity more available, double running is necessary, because the double moving average (DMA) successfully captures variations in velocity and, simultaneously, reduced scatter [11], as shown in Figure 11. The DMA can also eliminate the time lag effect on the original time series compared with the single MA [45].

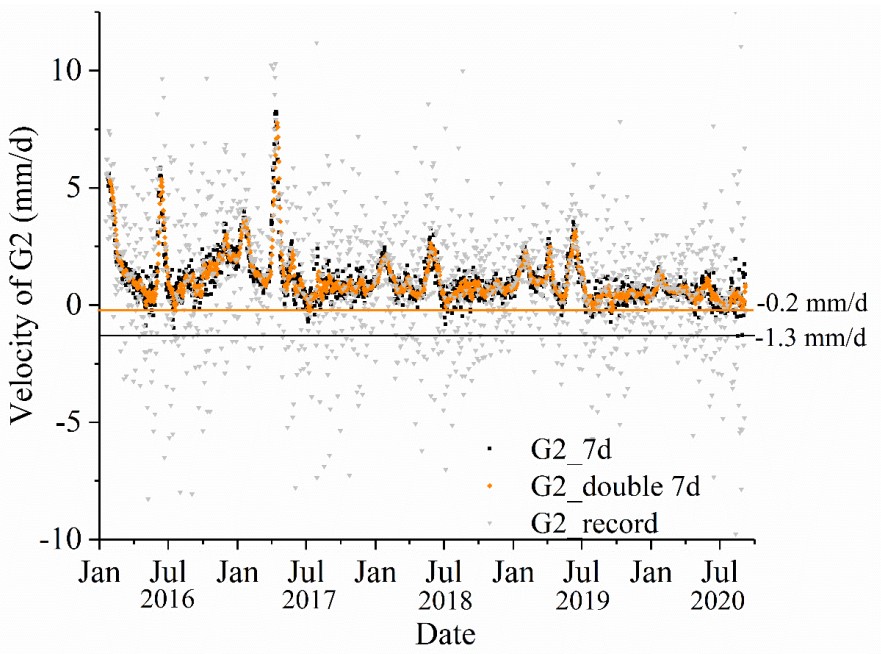

**Figure 11.** Original and MA/DMA velocity series of G2 unit during 2016~2020; −1.3 mm/d is the minimum MA velocity, and −0.2 mm/d is the minimum DMA velocity; the running window is 7 days.

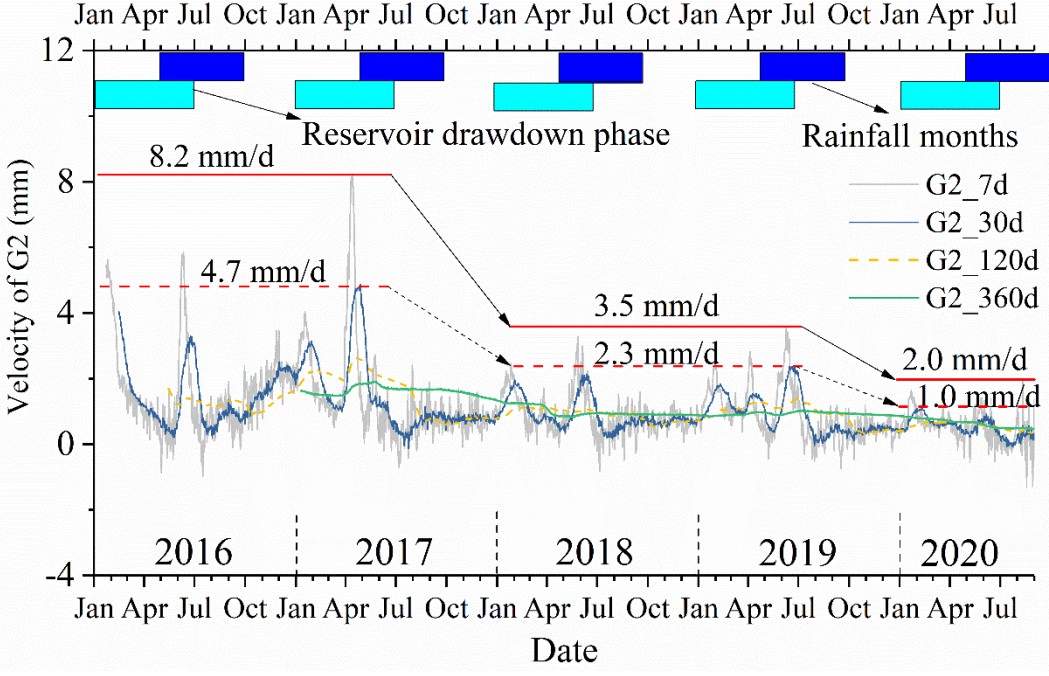

**Figure 12.** MA velocity series of G2 unit in short− and long−term running windows during 2016~2020. Seven (30) days are the short-term running window, and the yearly maximum MA velocities are 8.2 (4.7) mm/d in 2016 and 2017, 3.5 (2.3) mm/d in 2018 and 2019, and 2.0 (1.0) mm/d in 2020. In addition, 120 and 360 days are the long-term running windows, and maximum MA velocity is close to the mean in the velocity series.

The 8.2 mm/d velocity is the maximum weekly average velocity known to have occurred, and 2 mm/d is the maximum weekly average during the most recent fluctuation. Thus, all the velocity scatters are classified into velocity groups ranging from less than −8 mm/d to more than 8 mm/d to find the regularities of velocity distribution. The velocity groups between −8 and 8 mm/d are (−8, −2], (−2, 0], (0, 2], and (2, 8]. The statistics results of velocity in records, seven-day average, 30-day average, double seven-day average, double five-day average, and double three-day average are compared (Figure 13). The velocity group (0, 2] has the maximum number of frequencies in all situations, with percentages of 30.1%, 80.6%, 87.6%, 85%, 87.7%, and 86.4%, respectively. More than 80% of velocities in the running average are limited below 2 mm/d. Therefore, not only in the sixth fluctuation, the running average velocities are below 2 mm/d, but the majority of velocities are less than 2 mm/d. For the analysis result of G2, it seems that the running average velocity of 2 mm/d is an effective value for acceleration warning, but note that the velocity of more than 2 mm/d in original records has a percentage of 66.8%. It is not reasonable to directly use the threshold (2 mm/d) for warning in monitoring, as it brings too many false warnings.

In addition, in the cases of five-day and three-day double moving average velocities (Figure 13e,f), the percentages of measurement points below 0 mm/d are 4.9% and 10.8%, which are 2.1% and 8.0% higher than the seven-day double moving average, respectively. For maximizing the use of information from all data points in the double running average, the great majority of the points after the running average should be greater than zero to obtain a specific meaning. Based on this purpose, the smallest window for the short-term moving average is seven days in this study.

For the threshold values of the whole monitoring system, we pre-checked the Pearson correlation coefficient (PCC) of the cumulative displacement and velocity between each remaining GNSS unit and G2 unit. The results show that the displacement coefficients are more than 0.95 and the velocity coefficients can also exceed 0.8, both suggesting that the other GNSS units have a strong similarity with the G2 unit in variations of displacement and velocity. Therefore, the threshold value of 2 mm/d was tested for the G1, G4, G5, and G9 units. The triggering factor of the deformation velocity is the fluctuations of reservoir levels. Therefore, the time series, including velocity, water level, and rate of change in water level, are presented, featured in Figure 14. Taking the double seven-day running average results of G1 as an example (Figure 14a), running average velocity scatters are distributed in three areas. Most of them are less than 2 mm/d. Those above the 2 mm/d line concentrate on the areas right of the 1819 m line and left of the 1860 m line, respectively, which represents displacement acceleration. Taking the rate of water level change factor into account, accelerated behaviors occurred at the beginning of the reservoir drawdown (left of 1860 m line) and during the lower water level (right of 1819 m line); however, the rate of water level change correlates more poorly with the accelerated deformation than the effect of water level, as many colored points with a bigger filling rate or dropping rate are below the 2 mm/d line. The water level controls the accelerated deformation, instead of the water level change rate, which is similar to the Muyubao landslide in the TGRA [19].

Note that we have compared the results using the seven-day and 10-day running averages on the G2 monitoring site (Figure 14b,c). They have a similar three-area feature as described in G1, and the critical velocity of the two is the same. Thus, using a 10-day running window for MA would not obtain a lesser delimited velocity than the seven-day interval, but it requires more data and more monitoring time in practical application, which is contrary to the timeliness capability of EWS. Thus the seven-day is determined as the unique short-term running window for MA.

Besides the G1 and G2 units, similar findings can also be obtained by the rest of the GNSS points, but the difference is the onset of the monitoring time of these GNSS sites, which affects the highest running average velocity. G1 and G2 started monitoring in January 2016; G3 and G4 were recorded from March 2017; and G5 and G9 were installed in June 2018. The delimitation for velocities of the G5 site by the 2 mm/d line is higher than

reality (Figure 14d). However, if the G5 site had been monitored earlier, 2 mm/d would be proper. As the monitoring starts later, the acceleration area on the left of the 1860 m line gradually disappears, but the acceleration areas on the right of 1817–1819 m lines have been presented without exceptions. Below or above a critical water level is a criterion condition to judge the initiation or termination motions of many reservoir-induced landslides [55]. For all monitoring units, it can be concluded that the boundary of the acceleration area is delimited below the 1820 m water level. Thus, we selected a higher integer with a value of 1820 m as the critical level to unify the water level thresholds.

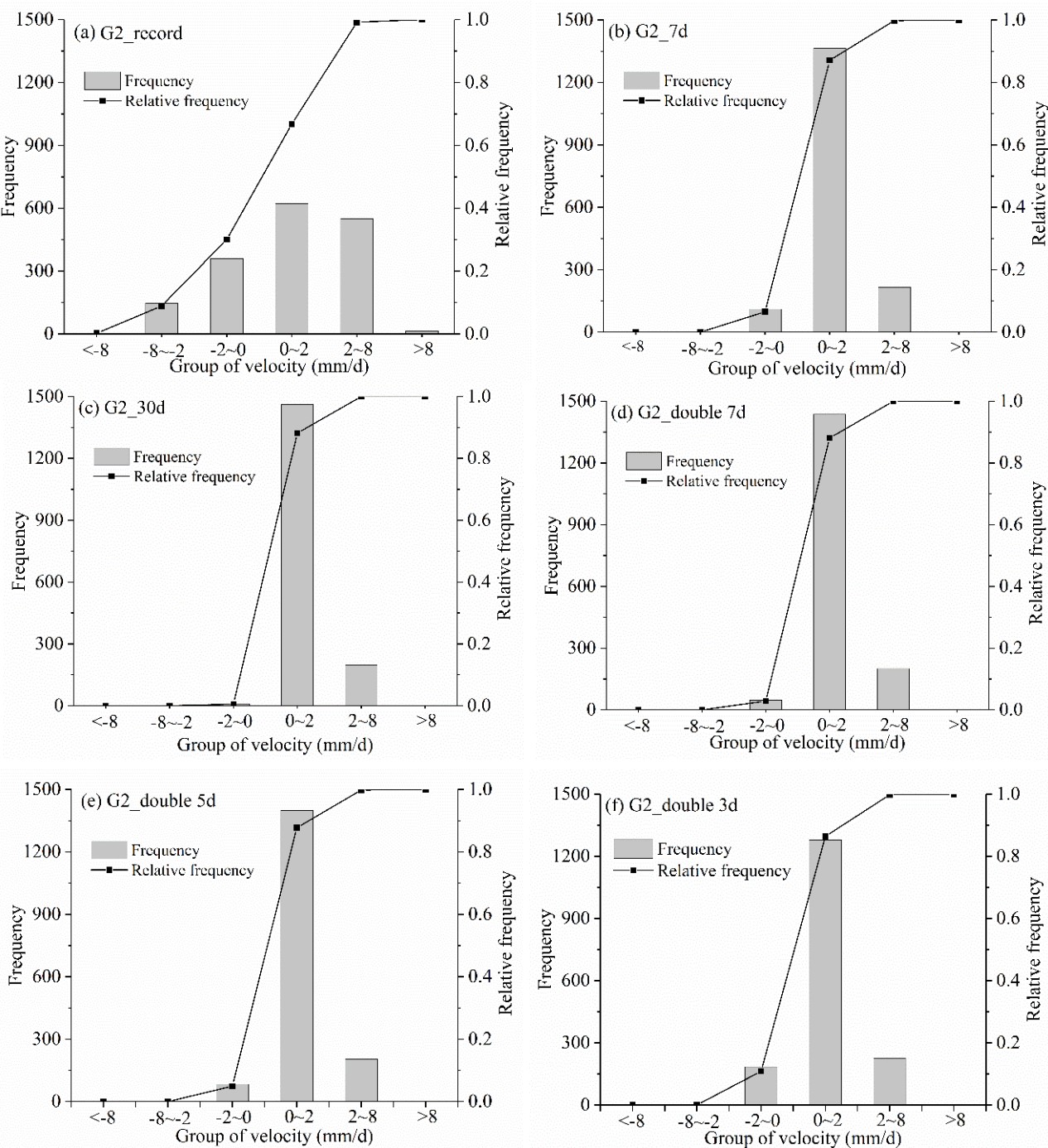

**Figure 13.** Histogram of rate distribution of G2: (**a**) recorded rates, (**b**) 7−day MA rates, (**c**) 30−day MA rates, (**d**) 7−day DMA rates, (**e**) 5−day DMA rates, (**f**) 3−day DMA rates.

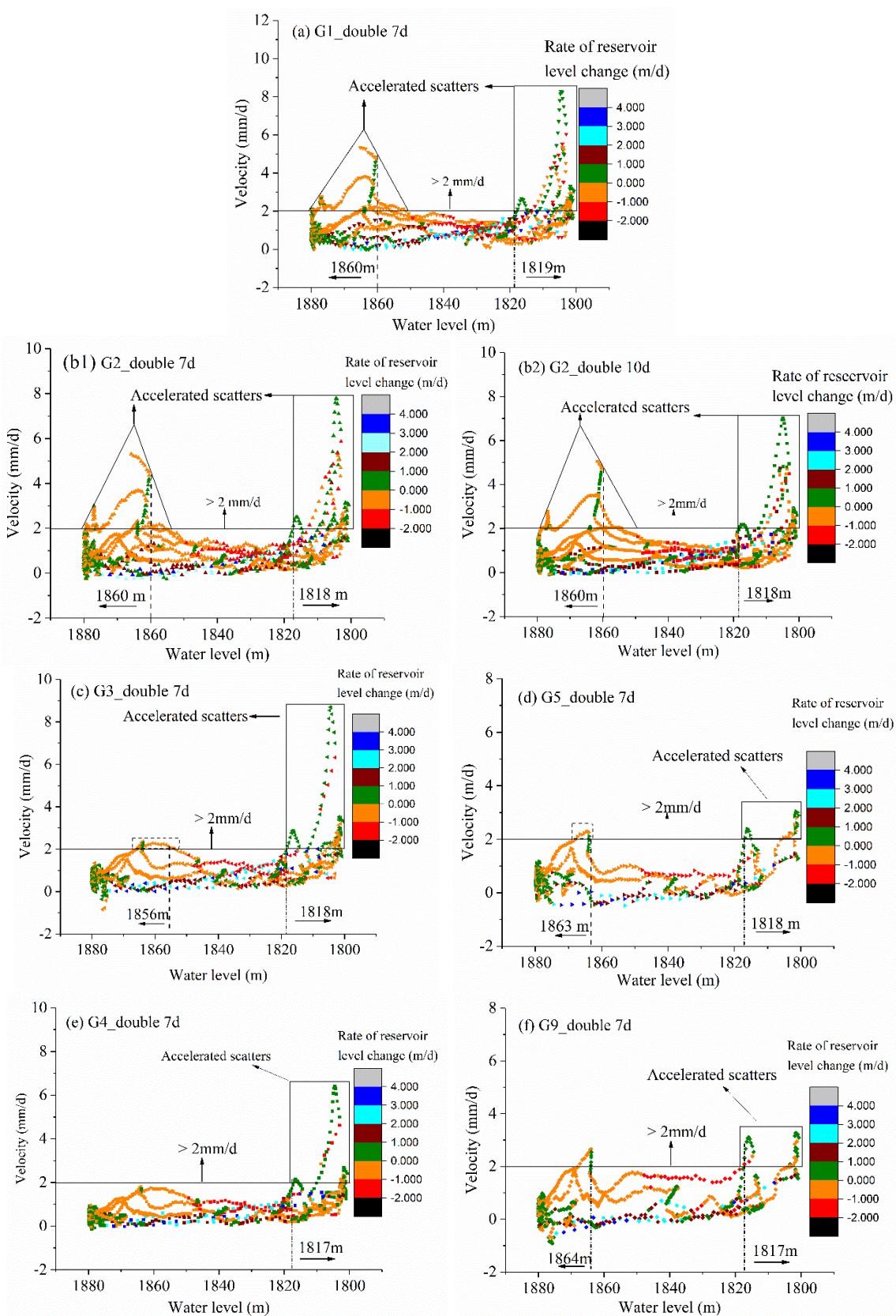

**Figure 14.** Correlations between the reservoir level and 7−day and 10−day DMA velocities: (**a**) G1_double 7-day, (**b1**) G2_double 7−day, (**b2**) G2_double 10−day, (**c**) G3_double 7−day, (**d**) G4_double 7−day, (**e**) G5_double 7−day, (**f**) G6_double 7−day. In these figures, scatters are colored by different levels of rate of reservoir level change, and the colored levels are shown at the right of every figure.

Water level below 1820 m is a dangerous scenario for the Gapa landslide, which is different to the abovementioned Muyubao landslide in the TGRA for which the warning water level is above a certain water level (172 m) [19]. The warning water level with opposite directions (below or above) of the two landslides can represent the seepage-induced and buoyancy-induced reservoir landslides, respectively, as indicated in the introduction section. Thus, the deformation mechanism controls the positive/negative threshold for critical water level.

In summary, two thresholds can be drawn from the displacement monitoring and reservoir levels: (1) the 2 mm/d of seven-day DMA velocity; and (2) the 1820 m water level. DMA velocity greater than 2 mm/d and water level below 1820 m asl are prone to trigger acceleration and be adverse to the Gapa landslide.

### 4.3. The Recommended Thresholds for EWS

In this case, Figure 11 has pointed out that the distribution of daily velocity data is extremely discrete with high noise. The daily recorded displacement velocity cannot be directly compared to the extracted threshold (2 mm/d), otherwise it would easily cause false warnings. For example, as Figure 13a shows, there are 564 velocity points greater than 2 mm/d, 33% overall over the G2 monitoring period. Therefore, a threshold for practical applications should be amplified to a certain extent, compared to the moving average threshold.

How can we reasonably and accurately redefine the abovementioned velocity threshold? Recognizing that the alarm should be triggered in an EWS's procedure if the preset thresholds are reached or exceeded is important. Accordingly, the maximum allowable rate in the EWS needs to be found; in other words, the displacement increment needs to be found as the lowest when reaching the DMA velocity of 2 mm/d by a running window. The smallest displacement increment means that the velocity series of the landslide reach the DMA threshold of 2 mm/d most easily, thus indicating that the maximum velocity in the smallest displacement increment is the maximum allowable velocity.

For this purpose, we assume that the time series of the double moving average velocity monotonically increases to 2 mm/d in 13 days. It is easy to find that this situation causes the smallest displacement increment if we set the starting point at zero in the linear/non-linear relationship. Within the non-linear relationship, this is because we can always set the starting point at zero and associate it with the endpoint and the local minimum value of the rate in the non-linear increase for time integrating. Thus, the velocity monotonically increasing with time obtains the smallest displacement increment among linear, concave, and convex curves.

Let us set $v_{13}''$ equal to 2 mm/d at time $t_{13}$ and have it linearly increased from $t_1$, as Figure 15 shows. According to Equations (1) and (2) and initial conditions ($v_{13}'' = 2, v_1 = 0$), and the divisions of every term used in the equations in Figure 15c, the value of $v_7$ can be mathematically derived to be equal to 2 mm/d. Giving $v_1$ equals zero, $v_{13}$ can be determined as a minimum of 4 mm/d. Similarly, in the case of a concave monotonic curve, $v_{13}$ is greater than 4 mm/d, whereas in the case of a convex monotonic curve, $v_{13}$ is less than 4 mm/d. However, daily velocity in the scatter plot is more concentrated on the onset of an acceleration episode, instead of later or final stages (Figure 11), which is more likely to be consistent with a convex curve. Consequently, 4 mm/d is recommended in an ideal way for the linear type, and, meanwhile, 4 mm/d is on the safe side for the convex type.

The velocity points beyond 4 mm/d are collected under two scenarios covering the monitoring period. Based on the recommended thresholds, scenario (1) solely meets recorded velocity >4 mm/d, and scenario (2) delimits that recorded velocity of >4 mm/d and a water level <1820 m. The number of warning times declines as the number of annual fluctuations increases (Figure 16). If the limiting condition (water level <1820 m) is combined, the number of warning times in scenario (2) sharply reduces compared to scenario (1). The annual relative frequency of all monitoring sites in scenario (2) is less than 10% and is almost less than 3% during the last three years. Since the original records are in

chaos in this reservoir-induced landslide case, the water level threshold can reduce false warning frequency. This is because velocity points above the 1820 m water level do not belong to the main accelerated zone (Figure 14).

The warning dates in scenario (2) during the last two years were gathered to analyze the temporal features of warnings. The water levels dropping to below 1820 m asl always happened in May and June, based on statistics. On the other hand, an episode that the water levels fill to 1820 m after dropping to 1800 m asl is also applicable to "water level <1820 m". It always finishes in a short episode (fewer than ten days) during late June to early July. The amount of data during this reservoir filling episode in June is quite small (about five each year), so the data collection in June is a mix of reservoir release and filling.

According to the warning times in May and June, as illustrated on the monthly calendars (Figure 17), the month of June 2019 had the most days of warning times. Over half of the monitoring units released warnings on 9, 13, 15, and 18 June 2019, on 22 and 24 May 2020, and on 14 June 2020; all units released warnings on 9 and 15 June 2019 and on 14 June 2020. From this point of view, the threshold is always discretely exceeded at present, and it is most frequently exceeded in June, especially in mid-June over the last two years.

If the thresholds of velocity and water level are both reached or exceeded, this means that short accelerations are likely to occur. Furthermore, if the thresholds are both continuously exceeded in monitoring days, the Gapa landslide may locally or completely enter the tertiary deformation. This suggests that at least upgrading the warning level to a "caution" level is necessary to allow dam operators to improve attention and surveillance on the landslide. Thus, the thresholds are efficient and effective for judging accelerations, otherwise accelerating behaviors can only be detected by the cumulative displacement, which takes more time and easily causes false warnings.

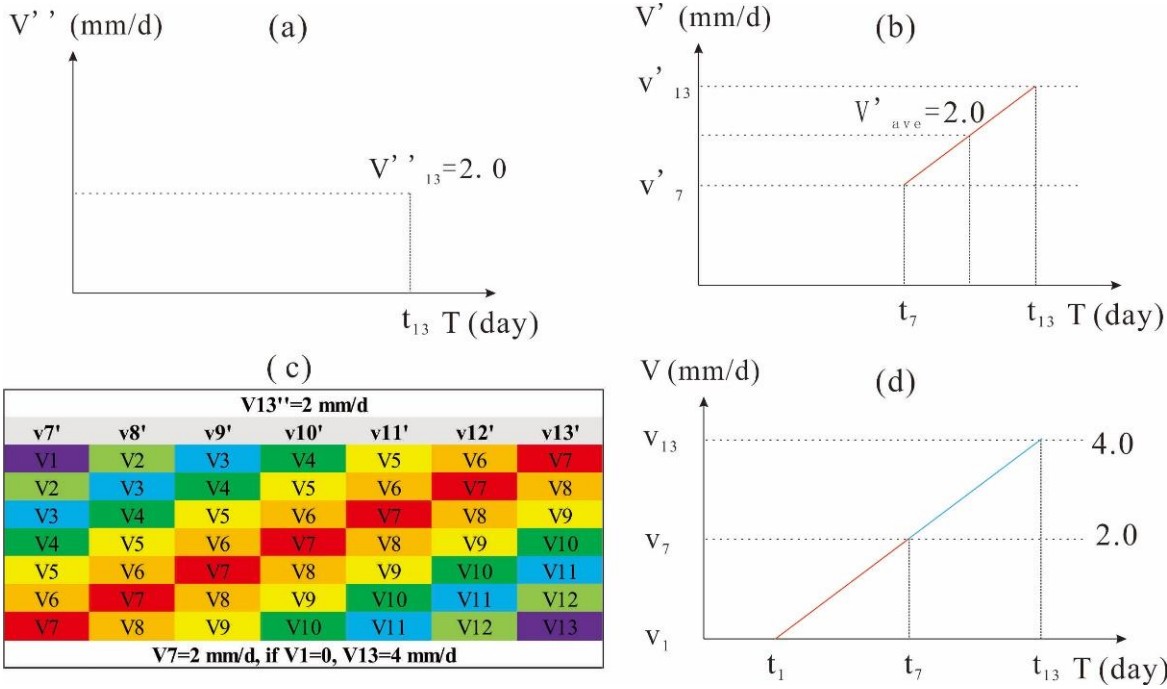

**Figure 15.** The process of defining a recommended velocity threshold using the reverse double moving average method in displacement vs. time in a linear relationship: (**a**) the DMA threshold plotted on DMA velocity vs. time; (**b**) the MA velocity series from $t_7$ to $t_{13}$ plotted on MA velocity vs. time; (**c**) the polynomial of every MA velocity from $t_7$ to $t_{13}$ expressed in columns (according to this table, the value of $V_7$ is found to equal 2 mm/d); and (**d**) the original velocity series from $t_1$ to $t_{13}$ plotted on velocity vs. time.

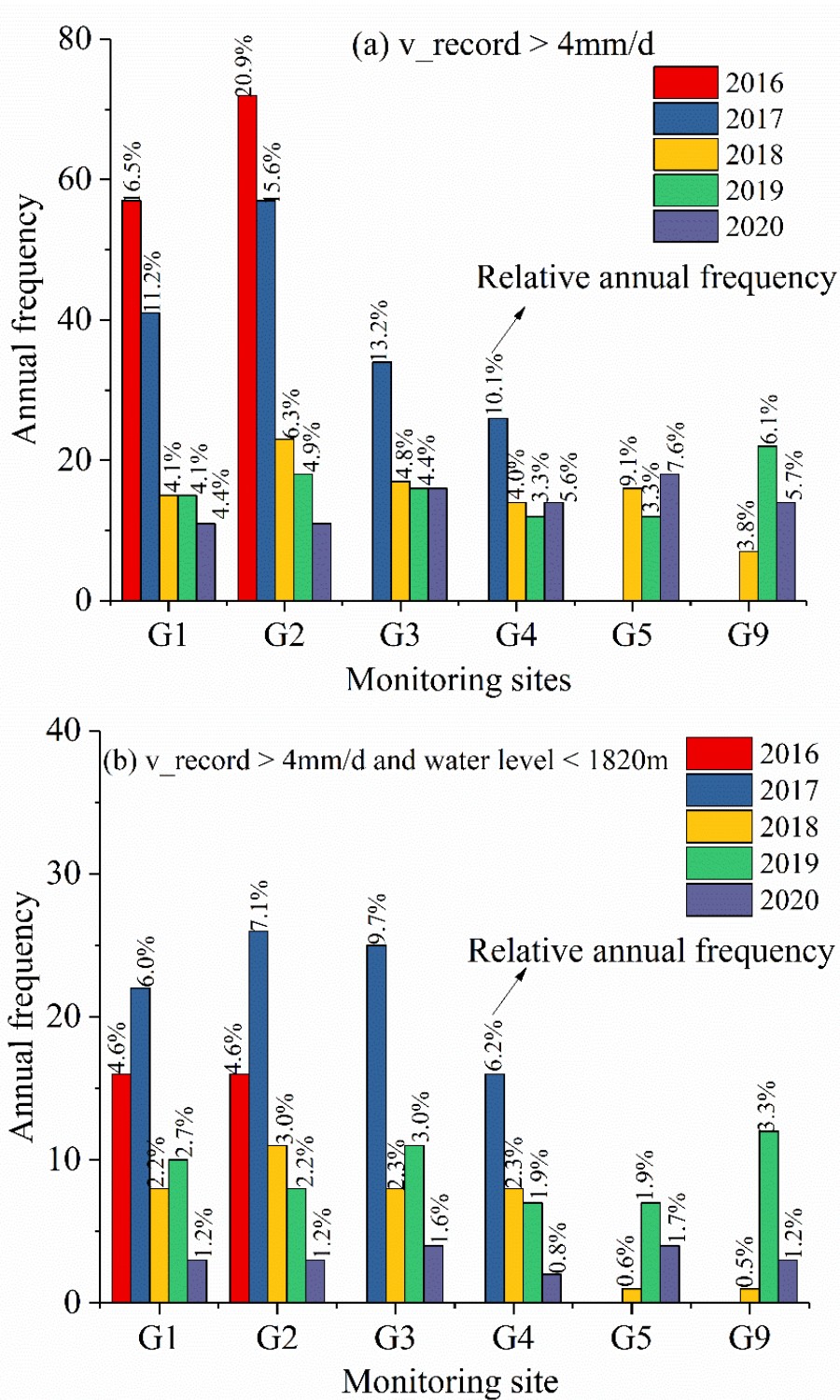

**Figure 16.** The frequency and annual relative frequency (given in percentage) exceeding the thresholds over monitoring years: (**a**) recorded daily velocity greater than 4 mm/d; (**b**) recorded daily velocity greater than 4 mm/d and water level less than 1820 m asl.

| Number of warning sites | | | | | | | Number of warning sites | | | | | | |
|---|---|---|---|---|---|---|---|---|---|---|---|---|---|
| 0 | 1 | 2 | 3 | 4 | 5 | 6 | 0 | 1 | 2 | 3 | 4 | 5 | 6 |
| May-19 | | | | | | | Jun-19 | | | | | | |
|  |  | 1 | 2 | 3 | 4 | 5 |  |  |  |  |  | 1 | 2 |
| 6 | 7 | 8 | 9 | 10 | 11 | 12 | 3 | 4 | 5 | 6 | 7 | 8 | 9 |
| 13 | 14 | 15 | 16 | 17 | 18 | 19 | 10 | 11 | 12 | 13 | 14 | 15 | 16 |
| 20 | 21 | 22 | 23 | 24 | 25 | 26 | 17 | 18 | 19 | 20 | 21 | 22 | 23 |
| 27 | 28 | 29 | 30 | 31 |  |  | 24 | 25 | 26 | 27 | 28 | 29 | 30 |
| May-20 | | | | | | | Jun-20 | | | | | | |
|  |  |  |  | 1 | 2 | 3 | 1 | 2 | 3 | 4 | 5 | 6 | 7 |
| 4 | 5 | 6 | 7 | 8 | 9 | 10 | 8 | 9 | 10 | 11 | 12 | 13 | 14 |
| 11 | 12 | 13 | 14 | 15 | 16 | 17 | 15 | 16 | 17 | 18 | 19 | 20 | 21 |
| 18 | 19 | 20 | 21 | 22 | 23 | 24 | 22 | 23 | 24 | 25 | 26 | 27 | 28 |
| 25 | 26 | 27 | 28 | 29 | 30 | 31 | 29 | 30 |  |  |  |  |  |

**Figure 17.** Early warning state in May and June 2019/2020. Number of GNSS units releasing warning spans from 0 to 6, 0 and 6 means none of the GNSS units to all GNSS units releasing warnings, respectively.

## 5. Discussion

Accelerations of the Gapa landslide movement often occur during reservoir fluctuation each year. Analysis of velocity time series based on the moving average method was conducted for warning of periodic acceleration phases with a velocity threshold. Meanwhile, a water level threshold was also assigned to reduce false warnings for GNSS units. The forward and reverse double moving average method is recommended for the time series of GNSS data to define thresholds in EWSs. The proposed method is beneficial in reducing the data noise level, extracting stable pre-thresholds, and defining a final threshold for better performance. Additionally, the water level threshold is crucial as well for warning of possible accelerations of the landslide, as it is mainly influenced by reservoir water fluctuations.

Although the current work provides a foundation for an EWS for the Gapa landslide, the improved Saito's model applied is still limited in different deformation stages, which means that it cannot release a time failure alarm for the landslide based on the current deformation stage. This is because the landslide is believed to enter or continue the constant deformation stage using existing GNSS units and data. Therefore, novel EWS models are urgently needed for landslides with periodic acceleration episodes. Another factor limiting the present EWS is that supplementing and installation of new GNSS units are not cost-effective (the cost of a single GNSS unit is close to 2000 EUR) [10]. The monitoring cost delays overall and detailed recognition of the landslide kinematics and, meanwhile, lowers the coving area of the EWS for the whole landslide. Some other remote sensing techniques could possibly be applied to this target. The Maoxian landslide occurred in June 2017 and killed more than 100 people. Intrieri et al. reported this case study using generated ground deformation maps and displacement time series for detecting precursors of failure with InSAR data [56]. Furthermore, techniques combining GNSS, satellite InSAR (interferometric synthetic aperture radar), ground-based InSAR, and aerial photogrammetry can enhance the understanding of large active landslides at the slope scale and the performance of EWSs, as two cases in Alpine and Apennines, Italy documented [57,58]. Therefore, InSAR techniques could potentially be employed in the Gapa landslide, because it is west-facing and has little effect of snow or vegetation coving in winter. Accessing InSAR data combined with GNSS units for the EWS is believed to enhance its scope and accuracy for the Gapa landslide. On the other hand, thresholds should be calibrated as soon as new data are available in order to guarantee that they are designed flexibly. The comprehensive development of an EWS with threshold update and InSAR technique supplements warrants future studies.

## 6. Conclusions

Synthesizing the above study, a method highlighted by forward and reverse moving averages for threshold definition is preliminarily proposed. The velocity and water level

thresholds were successfully applied to an EWS of the Gapa landslide. The conclusions can be drawn as follows:

(1) The Gapa landslide was formed by bending and topping between the late Pleistocene and early Holocene. The landslide had exhibited very slow movement for the decade before the reservoir impoundment. However, it was reactivated after the impoundment, and its movement was strongly related to the annual reservoir fluctuations, although the annual velocity has now gradually decreased during recent years. Two possibilities of the landslide's evolution—that it enters the acceleration stage or recovers from the periodic fluctuations to a slow-moving/stable state in the future—have been inferred. However, considering the uncertainty and limited monitoring period, the EWS for the Gapa landslide was developed using the improved Saito's model. The warning level is believed to be set at "attention level", according to the landslide's current motion.

(2) In this EWS, the velocity and water level are employed as combined warning indicators. The velocity threshold was defined by the forward and reverse moving average method, and, meanwhile, the seven-day window was decided as the short-term moving window. The recommended velocity threshold is 4 mm/d for current monitoring units. Additionally, a water level threshold with a specific elevation of 1820 m is also used for warning of possible accelerations, based on the distribution characteristics of the DMA velocity time series.

(3) If the daily recorded velocity in each monitoring site exceeds 4 mm/d and, simultaneously, the water level is below 1820 m elevation asl, the warning of likely accelerations will be released in this EWS, allowing users and managers to give more attention and surveillance to the potential hazard. Practical application showed that with the aid of the water level threshold, false warnings are effectively reduced, compared to the sole velocity threshold. It also pointed out that the most alert-prone time zone was mid-June during the last two years. As the thresholds were successfully defined and verified in this typical case, both the velocity and water level thresholds are recommended to be simultaneously used in EWSs for reservoir-induced large-scale landslides in southwestern China and other similar parts of the world.

**Author Contributions:** S.W. organized and analyzed the data and wrote the paper, X.H. supervised the work, W.Z. and M.B. reviewed and edited the manuscript, Z.Q. and W.S. validated landslide data. All authors have read and agreed to the published version of the manuscript.

**Funding:** This study was funded by the National Key Research and Development Program of China (No. 2017YFC1501302), the Key Program of the National Natural Science Foundation of China (No. 41630643), the Fundamental Research Funds for the Central Universities, and China University of Geosciences (Wuhan) (No. CUGCJ1701).

**Institutional Review Board Statement:** Not applicable for studies not involving humans or animals.

**Informed Consent Statement:** Not applicable for studies not involving humans.

**Data Availability Statement:** Restrictions apply to the availability of these data. Data were obtained from the Chengdu Engineering Corporation Limited and are available from the authors with the permission of the Chengdu Engineering Corporation Limited.

**Acknowledgments:** The first author appreciates the support for field monitoring and data provided by Yalong River Hydropower Development Company Limited and Chengdu Engineering Corporation Limited. He is also grateful to the China Scholarship Council for providing a scholarship for this research, which was conducted while he was a visiting PhD student (CSC202006410019) at the University of Bologna, Italy.

**Conflicts of Interest:** The authors declare no conflict of interest.

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
