# Peer review of "Threshold Definition for Monitoring Gapa Landslide under Large Variations in Reservoir Level Using GNSS"

_remotesensing, doi:10.3390/rs13244977_

Round 1

Reviewer 1 Report

The authors have derived an early warning system (EWS) for the Gapa Landslide adjacent to the reservoir behind the Jinping Dam on Yalong River in western China. The EWS threshold is based on both a running average of in situ GNSS displacement rates as well as reservoir water levels. In particular, the authors found that a displacement rate of 4 mm/day along with a reservoir water level below 1820 m would trigger their warning of impending acceleration.

In general, I think the paper is good. Some English editing is required as some of the talking points are a bit convoluted for a native English speaker. Also, I think the paper provides the reader with too many figures/much information. I'd suggest to only keep the most important figures in the main text, and the rest can be used in a supplementary section. Nearly all of the figures lack some corresponding text, requiring the reader to parse through the text to truly understand the figures themselves. I'd suggest adding in some accompanying information to the figure text (not just the figure title). I have added some more specific comments below. Further, the main results from this entire study depend on your GNSS measurements, but little information is actually provided about them. How were the processed? How much data was acquired per day? Etc. See one of the comments below. Lastly, I think a displacement rate threshold for each annual reservoir cycle type would be more beneficial. For example, the displacement rates are notably different during different times of the year. This is of course related to precip and water levels as well as reservoir operations. Your method now uses a double-MA technique to create the threshold. And this double-MA method essentially averages out your different displacement rates. I'd suggest creating a displacement method using your MA method for displacement rates during the seasonal reservoir filling period AND the seasonal reservoir release period. This could provide more meaningful and precise EWS to end-users. See one of the comments below.

Lines 36-38: It's not clear as to which reservoirs you're comparing water level height changes to the water level changes at TGR.

Lines 38-39: You bring up Yaxia dam here, but it is never mentioned again. If it's not pertinent for the story you want to tell in the introduction, then you shouldn't put that information in. This is true for all of your intro!

Line 46: What "intense" events are you referring to? E.g., Large precip events coincident with notable changes in reservoir water levels?? Or what?

Line 47-48: Your point about the need for EWS needs to be better defined as this is the basis of your work.

Line 50: " cost-effective and high efficiency" what? Or perhaps this an issue with language.

Line 54: Maybe provide an example of what you mean by "reginal" and "slope" scale.

Lines 57-60: Maybe some more sentences here on how these techniques are used to create EWS

Lines 63-64: Also slope and geology!

Lines 70-74: Maybe bring up the seepage-induced and buoyancy-induced types of landslides here? These types are mentioned farther down in the text.

Line 82: "environmental quantities" Such as?

Line 89-90: You bring up landslide "stages", but their isn't really background on that. I note you cite those studies above, but perhaps it's best to add a few sentences about the stages so that a reader will know what you mean here when you say "secondary stage" and acceleration" stage.

Line 107: Figure 1 is a good example of a figure that doesn't really need to be in the main text. It would be well-suited to a supplementary section.

Line 115-117: How did you determine that 97% of the 77 landslides are triggered/driven by reservoir impoundments?

Line 126: Figure 3 is another good example of a figure that doesn't really need to be in the main text. It would be well-suited to a supplementary section.

Line 135: How was the depth estimated?

Line 136: What is this "super-large riverside landslides" designation. Is it some sort of class of landslides? I don't think it's mentioned before here.

Line 140: What reservoir?

Line 141-142: There is no context here about what the river level elevation was before the impoundment..

Line 190: Figure 7b shows the accumulative displacement for GP2. Why is their negative accumulation for some years? And why do you think the rates are slowing down during these time periods?

Line 225: Maybe need to better describe the "fluctuation cycle". For example, make sure the reader understands that this a seasonal cycle is based on …

Line 245: Figure 9. This is a good example of why you should add more information to your figure captions. The reader doesn't know what the green/grey regions are at the toe of the slide..

Lines 304-314: I'm not convinced that you can conclude that the Gapa landslide will see deformation rates <= the recent seasonal rates based on a comparison of a few other reservoir-adjacent landslides. I think the Gapa rates are more of a function of precip and water level.

Line 315: Figure 12. It seems counterintuitive to me to have the alarm levels (IV to I) decrease as severity/displacement increases. Is this a known scale? It makes more sense to me for level IV to be more "serious" than level I.

Line 365: Figure 13. What do negative and positive displacement rates mean? Is there some sort directionality/angle of the displacement vector implied by the sign of the rate? Also, how are you deriving the initial rates (grey circles)? Euclidean distance between consecutive dates? What's the uncertainty in these measurements? How was the GNSS data processed (PPK, etc.)? How far away was the base station? The results from this entire study hinge on these GNSS measurements, but little information is provided about them.

Line 347: Figure 14. I think you'd get more meaningful information if you use your moving averages per reservoir cycle. For example, run the MA during the seasonal filling and also during the seasonal release. Otherwise, you're just averaging out the rates for the days that are with +- one half of the MA day value. Using this suggested method would also allow to create a meaningful threshold for the different cycles of the reservoir! Which, as you noted, is a driver for the displacement rates.

Author Response

Reply—The authors thank the reviewer for your constructive comments that significantly improved the quality of the manuscript. We have incorporated the comments into revising the manuscript accordingly. Notably, in the mentioned figure issue, the GNSS data processing, and the issue related to double-MA. The comments have been addressed one by one as the file attached. Also, please see the tracked changes in the manuscript.  

Reviewer 2 Report

The main goal of the research was to define algorithm for early warning system  (EWS) for the landslides in the proximity of the dams in China.

The authors  proved that GNSS observation of the ground movement and registration of water level changes may be the basis for EWS in the study area. The research are based on known modelling and monitoring methods, so the novelty of presented solution is rather moderate. The Satio model has been applied recently in many areas hazarded by landslides.

I think that presented research are interesting case study of joining the GNSS observation of displacement with water level changes in order to find threshold for the landslides.

  1. The advantage of the presented methodology is:

1.1. high accuracy of the observation of displacement,

1.2. potential of adjusting InSAR measurements to terrestrial measurements (the geodetic measurements should be done in the same period of the time when InSAR measurements were analyzed ). Based on recorded data back analysis could be done.

1.3. ability to analyze ground movements in any time.

1.4. good recognition of geological formation in the proximity of landsides.

1.5. Good record of the water level change, that support analysis of relation between landslides occurrence and water level changes.

  1. Disadvantages of presented method is

2.1. discreet character of the observation done by GNNS. This kind of the measurement are not cost effective.

2.2. lack of other remote sensing techniques (for example InSAR)

2.2. necessity of the adjusting InSAR measurements (if they were done) to terrestrial measurements

2.3. Modelling of landslide formation was carried out based on known Satio theory.

2.4. Lack of novelty in the presented solution

  1. Some editorial remarks

3.1.Please improve the quality of the:

-  figure 2 , 15 The font is to small,

3.2. Please try to unify the sizes and styles of the font on the figures (in general). The good explanation of the research (presented on the figures) is a strong point of the research.  Please try to work on that.

After improvement the article can be published

Author Response

Replies—The authors thank the reviewer for your constructive comments that significantly improved the quality of the manuscript. We have incorporated the comments into revising the manuscript accordingly. The comments have been addressed one by one as follows. Also, see the tracked changes of the manuscript.

The advantage of the presented methodology is:

1.1. high accuracy of the observation of displacement,

1.2. potential of adjusting InSAR measurements to terrestrial measurements (the geodetic measurements should be done in the same period of the time when InSAR measurements were analyzed). Based on recorded data back analysis could be done.

1.3. ability to analyze ground movements in any time.

1.4. good recognition of geological formation in the proximity of landsides.

1.5. Good record of the water level change, that support analysis of relation between landslides occurrence and water level changes.

Replies—Thanks for your all positive comments on the manuscript.

Disadvantages of presented method is

2.1. discreet character of the observation done by GNNS. This kind of the measurement are not cost effective.

Reply—Agreed. This is the main reseason that the GNSS units applied in this landslide are limited.

2.2. lack of other remote sensing techniques (for example InSAR)

Reply—Agreed. It is very meaningful to use other remote sensing techniques in the Gapa landslide, such as InSAR, the strong/weak deformation area is prone to be detected and the time series can be compared with GNSS data, especially, InSAR data can fill the displacement data lacking in 2015. In this regard, the authors have tried to access the data from InSAR for the recent half a year with some Italian and Chinese institutions. However, this work still has not yet progressed.

On the other hand, a discussion about the InSAR application in the Gapa landslide has been added. Please see Line 599-612. Anyway, the authors are looking forward to accessing InSAR data and doing related studies.

2.2. necessity of the adjusting InSAR measurements (if they were done) to terrestrial measurements

Reply—Agreed. But the work related to InSAR measurements has not been done yet.

2.3. Modelling of landslide formation was carried out based on known Satio theory.

Reply—Agreed. ‘Saito and Uezawa (1961) proposed the method based on the comparison between displacement records and creep rupture curves obtained from load controlled triaxial tests. Saito went on to present successful applications of this method in further studies (Saito, 1969, 1979)’ [1]. However, the creep feature may not suitable for the current and future deformation of the Gapa landslide. Because the Gapa landslide exhibits periodic accelerations induced by reservoir drawdown every year. The reservoir drawdown changes the stress state of the landslide, which disobeys the creep concept. Besides the Satio theory, the authors also investigated the Fukuzono model [2], also known as the inverse velocity method. But the results still needed to be validated and generalized.

References:

[1] Segalini A, Valletta A, Carri A. Landslide time-of-failure forecast and alert threshold assessment: A generalized criterion. Engineering geology, 2018, 245: 72-80.

[2] Fukuzono, T., 1985. A new method for predicting the failure time of a slope. In: Proceedings of the Fourth International Conference and Field Workshop on Landslides (Tokyo; 1985). 1985. Tokyo University Press, pp. 145–150.

2.4. Lack of novelty in the presented solution

Reply—Agreed. The procedure of the velocity definition is based on Saito’s model and the moving average method, although Saito’s model and moving average method are not original. But the authors not only have applied the model and method but also have improved them, for example, the forward and reverse double moving average method is proposed to define a velocity threshold due to the high noise level of the original velocity series. The method is beneficial in reducing the data noise level, extracting stable pre-threshold, and defining a final threshold for better performance. It has now been highlighted in the text. The application is believed to be effective for the EWS of the Gapa landslide during recent years.

Some editorial remarks

3.1.Please improve the quality of the:

- figure 2, 15 The font is to small,

Reply—Agreed. The font has been enlarged.

3.2. Please try to unify the sizes and styles of the font on the figures (in general). The good explanation of the research (presented on the figures) is a strong point of the research. Please try to work on that.

Reply—Corrected. The sizes and styles of the font on the figures have now been unified and figure text has been added to explain the information in detail of each figure. Notably, figure texts of Figures 2, 3, 8, 9, 11, 12, 14 and 15 have now been added to help the reader to parse through the text to truly understand the figures themselves.

After improvement the article can be published

Reply—Thanks for your constructive comments. The authors are very appreciative of that.

Reviewer 3 Report

Although the manuscript is overall interesting, it presents some issues that the Authors should address. All details are indicated in the following.

Required changes:

  1. The title seems to be too long and a shorter version should be chosen.
  2. Originality/novelty of the study proposed. This issue is very important and should be better clarified and well highlighted in the text.
  3. Quality of figures should be improved.
  4. Figures 13, 14 and 16: on the vertical axis, the term ‘displacement velocity’ should be replaced with ‘displacement’ or ‘velocity’.
  5. The early warning system is defined in this work on the basis of soil displacement. However, for large landslide bodies, the soil displacement might not be uniform in the entire soil mass. In this case, the velocities of different points of the slope could provide different levels of warning. This point should be adequately commented on in the manuscript.
  6. Introduction, lines 62-79: it would be worth mentioning other papers/criteria aimed to the prediction of threshold for landslide triggering. In this context, some suggested references are provided below.

SUGGESTED REFERENCES

Li L., Zhang S. X., Qiang Y., Zheng Z., Li S. H., Xia C. S. (2021). A Landslide Displacement Prediction Method with Iteration-Based Combined Strategy. Mathematical Problems in Engineering, vol. 2021, Article ID 6692503.

Huang F., Huang J., Jiang S., and Zhou C. (2017). Landslide displacement prediction based on multivariate chaotic model and extreme learning machine. Engineering Geology, 218, pp. 173–186, 2017.

Troncone A., Pugliese L., Lamanna G., Conte E. (2021). Prediction of rainfall-induced landslide movements in the presence of stabilizing piles. Engineering Geology 288, 106143.

Author Response

The authors thank the reviewer for your constructive comments that significantly improved the quality of the manuscript. We have incorporated the comments into revising the manuscript accordingly. The comments have been addressed one by one as the file attached.  Also, please see the tracked changes in the manuscript.  

Round 2

Reviewer 3 Report

The authors have properly addressed the comments raised during the previous round of review. Accondingly, the paper is suitable to be accepted in the current form.